# Poor Sensitivity of the MALDI Biotyper^®^ MBT Subtyping Module for Detection of *Klebsiella pneumoniae* Carbapenemase (KPC) in *Klebsiella* Species

**DOI:** 10.3390/antibiotics12091465

**Published:** 2023-09-20

**Authors:** Luz Cuello, Judith Alvarez Otero, Kerryl E. Greenwood-Quaintance, Liang Chen, Blake Hanson, Jinnethe Reyes, Lauren Komarow, Lizhao Ge, Zane D. Lancaster, Garrett G. Gordy, Audrey N. Schuetz, Robin Patel

**Affiliations:** 1Infectious Diseases Research Laboratory, Mayo Clinic, Rochester, MN 55905, USA; 2Hackensack Meridian Health Center for Discovery and Innovation, Nutley, NJ 07110, USA; 3Department of Epidemiology, Human Genetics & Environmental Sciences, School of Public Health, The University of Texas Health Science Center at Houston, Houston, TX 77030, USA; 4Molecular Genetics and Antimicrobial Resistance Unit, Universidad El Bosque, Bogotá 110121, Colombia; 5The Biostatistics Center, The George Washington University, Rockville, MD 20852, USA; 6Division of Clinical Microbiology, Department of Laboratory Medicine and Pathology, Mayo Clinic, Rochester, MN 55905, USA; 7Division of Public Health, Infectious Diseases and Occupational Medicine, Department of Medicine, Mayo Clinic, Rochester, MN 55905, USA

**Keywords:** carbapenem resistance, CRE, *Klebsiella*, KPC, MALDI-ToF mass spectrometry

## Abstract

Rapid detection of *Klebsiella pneumoniae* carbapenemase (KPC) in the *Klebsiella* species is desirable. The MALDI Biotyper^®^ MBT Subtyping Module (Bruker Daltonics) uses an algorithm that detects a peak at ~11,109 m/z corresponding to a protein encoded by the *p019* gene to detect KPC simultaneously with organism identification by a matrix-assisted laser desorption ionization–time-of-flight mass spectrometry (MALDI-ToF MS). Here, the subtyping module was evaluated using 795 clinical *Klebsiella* isolates, with whole genome sequences used to assess for *bla*_KPC_ and *p019*. For the isolates identified as KPC positive by sequencing, the overall sensitivity of the MALDI-ToF MS subtyping module was 239/574 (42%) with 100% specificity. For the isolates harboring *p019*, the subtyping module showed a sensitivity of 97% (239/246) and a specificity of 100%. The subtyping module had poor sensitivity for the detection of *bla*_KPC_-positive *Klebsiella* isolates, albeit exhibiting excellent specificity. The poor sensitivity was a result of *p019* being present in only 43% of the *bla*_KPC_-positive *Klebsiella* isolates.

## 1. Introduction

Antimicrobial resistance (AMR) is an increasingly pressing challenge. A comprehensive study performed by the Antimicrobial Resistance Collaborators estimated that globally in 2019, 4.95 and 1.27 million deaths were associated with or directly attributable to AMR, respectively [1]. As the leading AMR-types, carbapenem-resistant Enterobacterales (CRE) are classified as urgent threats by the United States Center for Disease Control and Prevention (CDC), with *Klebsiella pneumoniae* being a common carbapenem-resistant species [2].

Carbapenems have broad antimicrobial activity against Gram-negative and Gram-positive bacteria [3]. Carbapenem-resistant strains may be associated with multidrug resistance, severe infections, and poor treatment outcomes [4]. Although Enterobacterales may acquire carbapenem resistance by different mechanisms, *Klebsiella pneumoniae* carbapenemase (KPC) is a common enzyme conferring resistance to carbapenems in Enterobacterales and can confer resistance in non-fermenting Gram-negative bacilli such as *Pseudomonas aeruginosa* [5,6]. According to Ambler’s scheme, KPCs belong to the class A β-lactamase family of enzymes, containing a serine residue at their active site that allows for hydrolysis of the β-lactam ring of aztreonam, carbapenems, cephalosporins, and penicillins [7,8]. Variants of the *bla*_KPC_ gene occur and can be found on plasmids, often in close association with mobile elements [9,10], or on the chromosome. The former accounts for the effectiveness of KPCs in spreading between strains, species, and genera, posing a major challenge, particularly in hospital settings. Accordingly, there is a need for methods that rapidly detect KPC-harboring bacteria suitable for use in the clinical microbiology laboratory to quickly inform treatment options.

Matrix-assisted laser desorption ionization–time-of-flight mass spectrometry (MALDI-ToF MS) has been incorporated into most clinical microbiology laboratories in many countries for the rapid and reliable identification of bacteria grown in culture [11,12]. MALDI-ToF MS is a soft ionization method where matrix-embedded whole-cell-associated microbial analytes are desorbed by a laser [11,13]. The ionized microbial analytes are then accelerated based on their mass-to-charge (m/z) ratio through a vacuum in a time-of-flight (ToF) analysis, with analytes of low mass traveling faster and arriving at a detector ahead of the larger ones [11,13]. This process generates a readout of a mass spectrum unique to most individual bacterial species, allowing for the identification of species by comparison to a database of spectra with a fast turnaround time and reasonable cost [11]. Thus, MALDI-ToF MS has become a powerful tool for the quick, specific, and relatively inexpensive identification of bacteria (and fungi) in clinical laboratories [12]. MALDI-ToF MS is not, however, commonly used to characterize antimicrobial susceptibility. This has resulted in species identification being available well ahead of results for antimicrobial susceptibility testing in most instances. The delay in availability for antimicrobial susceptibility testing results leaves some patients “over” treated and others “under” treated, depending on the empirically selected antibiotic regimen and the susceptibility of the bacterium with which individual patients are infected.

In 2014, a study by Lau and collaborators reported a ~11,109 m/z peak in *K. pneumoniae* clinical isolates, associated with a *bla*_KPC_-bearing pKpQIL plasmid; it was suggested that the detection of this peak might be used to identify the KPC-harboring *Klebsiella* species [14]. The peak corresponds to a cleavage product of P019, a hypothetical protein encoded by the *p019* gene on the pKpQIL plasmid [14,15]. Prior studies have assessed the sensitivity and specificity of interrogating the ~11,109 m/z peak using automated detection algorithms coupled with MALDI-ToF MS to characterize KPC positivity with varying results. Cordovana and colleagues reported a KPC detection sensitivity and specificity of 85 and 100%, respectively, in 6209 *K. pneumoniae* isolates from Italy and Germany (including 2390 KPC producers), using the commercial MALDI Biotyper RUO and Flex Analysis 3.4 software (Bruker Daltonics, Billerica, MA, USA) [16]. Gaibani and colleagues tested the MALDI Biotyper RTC subtyping module (Bruker Daltonics) using 59 *K. pneumoniae* isolates (including 34 KPC producers), reporting a sensitivity of 98% and specificity of 99% [17]. Youn and colleagues evaluated 140 Enterobacterales (60 KPC producers, 26 with the *p019* sequence) and reported a sensitivity and specificity of 96 and 99%, respectively, for *p019*-harboring isolates using the Bruker BioTyper and Flex Analysis software version 3.4 (Bruker Daltonics, Billerica, MA, USA) [18]. Gato and colleagues performed a visual analysis of the mass spectral peaks obtained with the Bruker Daltonics subtyping module (Flex Control software 3.4 and Compass software 4.1.100) using 435 *K. pneumoniae* isolates (65 KPC producers), reporting a sensitivity of 49% and specificity of 100% [19].

The MALDI Biotyper^®^ MBT Subtyping Module (Bruker Daltonics) characterizes *Citrobacter freundii*, *Enterobacter aerogenes*, *Enterobacter asburiae*, *Enterobacter cloacae*, *Enterobacter kobei*, *Enterobacter ludwigii*, *Escherichia coli*, *Klebsiella aerogenes*, *Klebsiella oxytoca*, *Klebsiella pneumoniae*, *Klebsiella variicola*, and *Serratia marcescens* as “KPC positive” if the mass spectral analysis produces a peak at ~11,109 m/z [20]. To expand on the previous work and further define the sensitivity and specificity of automated KPC detection by a commercially available software tool, we tested the performance of KPC detection by the MALDI Biotyper^®^ MBT Subtyping Module using 795 clinical Antibacterial Resistance Leadership Group (ARLG) *Klebsiella* isolates from the Consortium on Resistance Against Carbapenems in *Klebsiella* and other Enterobacterales (CRACKLE-2) study [21,22]. The mass spectrometry results were compared to those of whole genome sequencing (WGS) to further interpret the proteomics results in the genetic context of the isolates analyzed.

## 2. Results

### 2.1. Characterization of Isolates by WGS

In total, 795 clinical ARLG *Klebsiella* isolates from CRACKLE-2 were tested, 792 (99.6%) of which were *K. pneumoniae*, with the remaining 3 (0.4%) being *K. variicola*. The species identification by MALDI-ToF MS matched the results for identification by WGS for all isolates. Geographically, the isolates were from four regions: (i) United States of America (USA), (ii) Central/Northern South America (Colombia and Nicaragua), (iii) Southern South America (Argentina and Chile), and (iv) Asia–Pacific (Lebanon and Singapore), although most isolates (739/795, 93%) were from the USA. All *K. variicola* isolates were from the USA, and 1/3 (33%) harbored the *bla*_KPC3_ gene (Figure 1a).

Among the *K. pneumoniae* isolates, the *bla*_KPC2_ gene was identified in isolates from the USA (320/736, 44%) and Central/Northern South America (1/6, 17%). Furthermore, 34% (248/736) of *K. pneumoniae* isolates from the USA harbored *bla*_KPC3_, while 0.5% (4/736) harbored other KPC genes. The isolates from Southern South America and Asia–Pacific were all *K. pneumoniae,* and none harbored *bla_KPC_* (Figure 1a,b).

### 2.2. KPC Detection by WGS and MALDI-ToF MS

The analysis of the results obtained by WGS showed that of 795 isolates tested, 574 (72%) were KPC-positive. Among the KPC-positive isolates, 321/574 (56%) harbored *bla*_KPC2_, 249/574 (43%) *bla*_KPC3_, and 4/574 (1%) other KPC subtypes (*bla*_KPC8_, *bla*_KPC28_, *bla*_KPC31_, or *bla*_KPC34_). The Biotyper^®^ MBT Subtyping Module identified 239/795 (30%) as KPC-positive (Table 1). Considering the number of isolates identified as KPC-positive by WGS, the overall sensitivity of the MALDI-ToF MS subtyping module was 239/574 (42%); the specificity was 100%.

Since the Biotyper^®^ MBT Subtyping Module relies on the detection of a ~11,109 m/z peak associated with the P019 protein for KPC identification, the presence of the *p019* sequence was assessed. The WGS data revealed that 246/795 isolates (31%) harbored the *p019* sequence. Moreover, of the 574 isolates identified as *bla*_KPC_ positive, 222/574 (39%) harbored *bla*_KPC2_ and *p019*, and 24/574 (4%) *bla*_KPC3_ and *p019* (Table 2). As shown in Table 2, the presence of *bla*_KPC2_ and *bla*_KPC3_ was not always associated with *p019*; none of the isolates harboring other KPC subtypes carried the *p019* gene.

The sensitivity of the subtyping module among isolates harboring *p019* was also assessed. Of the 246 isolates containing *p019*, the MALDI-ToF subtyping module typed 239 as KPC-positive, for a sensitivity of 97% and a specificity of 100% (Table 3).

During the proteomics study, the isolates were tested in duplicate. Notably, 22 MALDI-ToF MS plate spots showed discrepancies in KPC detection; upon reshooting, 11/22 (50%) were resolved. Similarly, 14 spots were reshot due to initially yielding an identification score <2.0; upon reshooting, 5/14 (36%) spots were resolved (identification score ≥ 2.0). All isolates yielded at least one identification spot score ≥ 2.0. 

## 3. Discussion

In this study, the Biotyper^®^ MBT Subtyping Module correctly identified 239/574 KPC-positive *Klebsiella* isolates, for an overall sensitivity of 42% compared to WGS, with excellent specificity. This poor sensitivity makes KPC detection via MALDI-ToF MS using the Biotyper^®^ MBT Subtyping Module unreliable for use in clinical laboratories. Dual spotting was performed, and any isolate with at least one spot positive according to MADLI-ToF MS was reported as KPC-positive. Although dual spotting may have increased the sensitivity of KPC detection, it adds work in routine clinical practice. It should be noted that the module assessed is not cleared by the United States Food and Drug Administration [20]. The results of this study show that although the Biotyper^®^ MBT Subtyping Module exhibits excellent specificity, the sensitivity is low, and further testing of the isolates is necessary to confirm KPC absence.

KPC identification by the Biotyper^®^ MBT Subtyping Module is based on the detection of an ~11,109 m/z mass spectral peak, provided that a successful identification score (log score ID ≥ 2.0) is obtained [20]. This peak corresponds to an abundant hypothetical protein termed P019, encoded by the *p019* gene in the pKpQIL plasmid that also harbors *bla*_KPC_. Thus, the detection of P019 by mass spectrometry is used as a proxy for KPC. In this study, the identification of KPC in isolates that did not harbor the *p019* sequence failed even with successful species identification. Unsurprisingly, there was a higher detection sensitivity for the subtyping module for isolates harboring the *p019* sequence (97%). Collectively, these results show that the poor sensitivity of the Biotyper^®^ MBT Subtyping Module was a result of the low representation of the *p019* sequence (*n* = 246/574, 43%) among the *bla*_KPC_-positive *Klebsiella* isolates studied. Although the ~11,109 m/z peak correlates with the presence of KPC, the absence of this peak does not imply the absence of KPC. Previous observations show that *bla*_KPC_ is not always associated with pKpQIL or *p019*, therefore *bla*_KPC_ and *p019* may be found independently [15,19]. 

In addition to assessing automated detection, the mass spectra were visually inspected. The assessors were blinded to the results of automated detection. Qualitatively, a high degree of variability among mass spectra was apparent. Typically, the isolates identified as KPC-positive using the software showed a clear peak at ~11,109 m/z, but sometimes the peak was surrounded by noise. It was unclear to what extent noise around the peak hindered KPC detection or above what threshold intensity value the ~11,109 m/z peak was successfully detected by the software. Moreover, discrepancies between spots of the same isolate were observed, even when both spots yielded successful identification scores (≥2.0). In these cases, reshooting the KPC-undetected spot resolved detection in 11/22 (50%) isolates. However, as discussed above, dual spotting would increase the workload in clinical settings, and in settings where dual spotting is not performed, there is a higher risk of KPC underdetection.

A limitation of this study is the geographical and species distribution of the isolates tested, as well as the large number of *K. pneumoniae* compared to *K. variicola* isolates. Although a large number of clinical isolates was examined, a majority were *K. pneumoniae* and were from the USA. Future studies could assess performance of the Biotyper^®^ MBT Subtyping Module in other KPC-positive species from a wider range of geographical locations. 

The escalating AMR represents a threat for human, animal, and environmental health, with profound socio-economic impacts associated with mortality and disability [2,4,23]. The rapid spread of resistance heightens the need for fast, accurate, and reliable detection methods that can guide treatment and reduce hospital-associated costs. Due to its simplicity and the possibility of obtaining accurate results in a fast and relatively inexpensive manner, MALDI-ToF MS is a gold-standard method for the identification of bacteria and fungi [11,12,24]. However, the automated detection of resistance mechanisms, such as KPC, by means of MALDI-ToF MS remains a challenge. The results of this study support the findings of Gato and colleagues [19], showing low sensitivity of the Biotyper^®^ MBT Subtyping Module in KPC isolates that do not harbor the *p019* sequence. The correlation between the P019 cleavage product detected by MALDI-ToF MS and the presence of *bla*_KPC_ is imperfect, making the use of this surrogate biomarker a poor way of detecting KPC [15,25]. While the automated detection of AMR mechanisms by MALDI-ToF MS would revolutionize workflow in the clinical microbiology laboratory and provide crucial information for ideal treatment, the indirect detection of KPC by the Biotyper^®^ MBT Subtyping Module is unreliable, failing for isolates that do not carry the *p019* sequence but harbor *bla*_KPC_. Local epidemiology and patterns of resistance-associated plasmid transmission should be considered when interpreting KPC detection by MALDI-ToF MS.

The use of MALDI-ToF MS for the automated detection of AMR mechanisms has enormous potential but requires further development. A possible improvement could be the identification of other high intensity peaks in mass spectra that may constitute potential biomarkers for resistance, such as the 4521 m/z peak identified in *bla*_KPC2_-bearing *K. pneumoniae* clinical isolates from China by Huang and colleagues [26]. However, the identification of novel biomarkers needs a thorough understanding of their genetic context to verify their linkage to resistance genes. Another interesting approach was recently published by Gato and colleagues, who proposed the use of the full spectrum of the bacterial proteome coupled with machine learning to predict OXA-48 and KPC carriage [27]. While the results of this study need external confirmation and evaluation using a more epidemiologically diverse set of isolates, the possibility presents an exciting avenue for the application of MALDI-ToF MS for the rapid and reliable clinical detection of AMR.

## 4. Materials and Methods

### 4.1. Subculturing Isolates and Control Bacterial Strains

The bacterial isolates were selected based on the availability of *Klebsiella* isolates and associated sequencing data at the time of the study. They constitute a subset of CRACKLE-2 isolates from previous studies [21,22]. First, 795 ARLG *Klebsiella* isolates (including *K. pneumoniae* and *K. variicola*) were subcultured on 5% sheep blood agar and incubated overnight at 37 °C. BAA 1705 (*K. pneumoniae*, *bla*_KPC_-positive isolate) and BAA 1706 (*K. pneumoniae*, *bla*_KPC_-negative isolate) were used as the positive and negative controls, respectively, and subcultured daily from frozen vials on 5% sheep blood agar and incubated overnight at 37 °C. ATCC 25923 (*Staphylococcus aureus*) was used as a control for species identification and was provided by the Clinical Bacteriology laboratory at Mayo Clinic, Rochester, MN, USA.

### 4.2. MALDI-ToF MS Plate Preparation

Spots on the MALDI-ToF MS plates were inoculated with 1 μL of 70% formic acid. Since the intensity of the KPC-related peak can be low [16], the subcultured colonies were tested in duplicate; colonies were transferred into the formic acid and allowed to air-dry. Single colonies were generally picked, with rare exceptions where colonies were small and more than one needed to be picked. Then, 2 μL of matrix solution (α-cyano-4-hydroxycinnamic acid HCCA) was overlaid onto each spot and allowed to air dry. The first and the last spot were reserved for calibration and loaded with bacterial test standard (BTS, Bruker Daltonics). Additionally, all of the plates included one spot inoculated with ATCC 25923 (positive control for species identification), one spot inoculated with BAA 1705 (*bla*_KPC_-positive control), one spot inoculated with BAA 1706 (*bla*_KPC_-negative control), and one spot with only 1 μL of 70% formic acid and 2 μL of the matrix solution (negative loading control). Since reusable stainless-steel plates were used, the negative loading control was spotted in a different location on the MALDI-ToF MS plate each day (to assess plate cleaning).

### 4.3. Obtaining MALDI-ToF MS Readouts

The MALDI-ToF MS plates were placed into the MALDI Biotyper^®^ Sirius system and calibration was performed on the BTS spot. The mass spectral readouts were obtained using Compass 4.1.100 and FlexControl 3.4.206 software.

As per the manufacturer’s suggestions [20], subtyping relied on successful species identification (log score identification ≥ 2.0). Thus, the individual spots were reshot when: (a) the identification score was <2.0; (b) duplicate spots yielded discrepant results regarding KPC detection; or (c) no peaks were found. Appendix A shows the criteria used for reshooting isolates and how the results were recorded. Entire plates were re-loaded when either a positive or negative control failed twice. Cordovana and colleagues observed no difference in KPC-detection between isolates cultured for 24, 48, or 76 h [16]. Thus, for this study, isolates that needed to be reloaded on a MALDI-ToF MS plate due to the failed identification of controls, were reloaded after a maximum of 48 h.

The mass spectral readouts were visually examined for the presence of the ~11,109 m/z peak. The evaluators were blinded to the automated detection results generated by the subtyping module.

### 4.4. Statistical Analysis and Preparation of Figures

The isolates were recorded as KPC-positive by MALDI-ToF MS if at least one spot was classified as positive by the Biotyper^®^ MBT Subtyping Module. During proteomic studies, the personnel were blinded to the WGS data (previous data [21,22]). Only after all isolates were processed by mass spectrometry and the results were analyzed were findings compared to those of WGS. Sensitivity was calculated as the percentage of the isolates detected as KPC-positive by MALDI-ToF MS among those positive by WGS. Specificity was calculated as the inverse of KPC-negative isolates by WGS that were detected as KPC positive by MALDI-ToF MS. The figures were created using Excel, Power Point, and MapChart. The graphical abstract and Appendix A were created with BioRender (https://www.biorender.com/, accessed on 25 August 2023).

### 4.5. Whole Genome Sequencing

The WGS data used for comparison to the MALDI-ToF MS results in this study were previously generated [21,22] at the following locations: Molecular Resource Facility, Rutgers, New Brunswick, NJ, USA (Illumina NextSeq 500; Illumina, Inc., San Diego, CA, USA); UTHealth, Houston, TX, USA (Illumina MiSeq; Illumina, Inc., San Diego, CA, USA); Baylor College of Medicine, Houston, TX, USA (Illumina HiSeq X; Illumina, Inc., San Diego, CA, USA); and Universidad El Bosque, Bogotá, Colombia (MiSeq, HiSeq 4000, and NextSeq 2000; Illumina, Inc., San Diego, CA, USA). The presence of the *p019* gene was assessed using BLASTN against the *Klebsiella* genomes from CRACKLE-2.

## Figures and Tables

**Figure 1 antibiotics-12-01465-f001:**
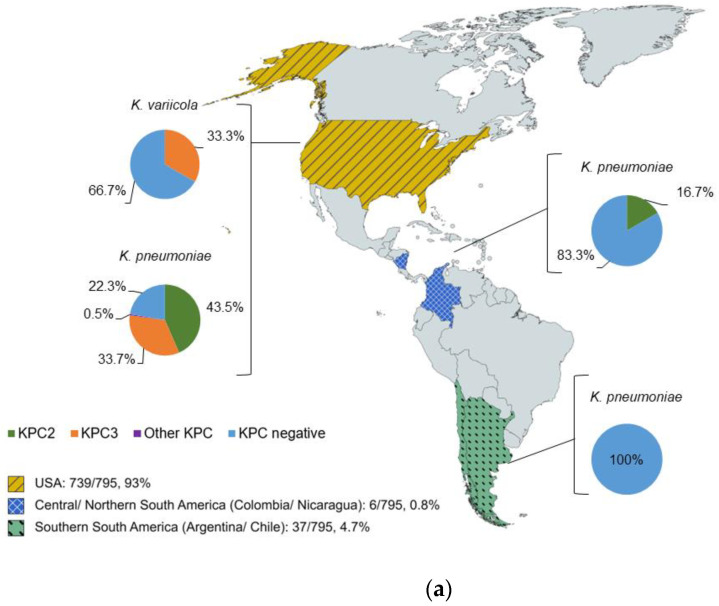
(**a**) The geographical distribution of *bla*_KPC_ genes in *Klebsiella* isolates in the USA, Central/Northern South America, and Southern South America regions. (**b**) The geographical distribution of *bla*_KPC_ genes in *Klebsiella* isolates in the Asia–Pacific region.

**Table 1 antibiotics-12-01465-t001:** Comparison of KPC status, as determined by WGS or MALDI-ToF MS.

WGS		MALDI-ToF MS	
	Number		Number
*bla*_KPC_ positive	574	KPC-positive	239
*bla* _KPC2_	321		
*bla* _KPC3_	249		
Other *bla*_KPC_	4		
*bla*_KPC_ negative	221	KPC negative	556

Of the 239 identified as KPC-positive by the MALDI-ToF MS subtyping module, 219 harbored *bla*_KPC2_ and 20 harbored *bla*_KPC3_.

**Table 2 antibiotics-12-01465-t002:** Distribution of the *p019* sequence in the 574 KPC-positive isolates, as determined by WGS.

	*p019*-Positive		*p019*-Negative	
	Number	Percent	Number	Percent
*bla* _KPC2_	222	39	99	17
*bla* _KPC3_	24	4	225	39
Other *bla*_KPC_	0	0	4	0.7
Total	246	43	328	57

**Table 3 antibiotics-12-01465-t003:** MALDI-ToF MS KPC status according to presence or absence of the *p019* sequence, as determined by WGS.

		WGS			
		*p019*-Positive		*p019*-Negative	
		Number	Percent	Number	Percent
MALDI-ToF MS	KPC-positive	239	30	0	0
	KPC-negative	7	1	549	69
Total		246	31	549	69

## Data Availability

The WGS data used for this study were previously generated [21,22] and can be found under https://doi.org/10.1016/S1473-3099(19)30755-8 and https://doi.org/10.1016/S1473-3099(21)00399-6.

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
