# Peer review of "Poor Sensitivity of the MALDI Biotyper® MBT Subtyping Module for Detection of Klebsiella pneumoniae Carbapenemase (KPC) in Klebsiella Species"

_antibiotics, 2023, doi:10.3390/antibiotics12091465_

Round 1
Reviewer 1 Report
This article is very interesting, very well written and works with a very important topic in clinical microbiology that is quick detection of KPC gene using MBT Subtyping Module . Although the overall sensitivity of the MALDI-ToF MS subtyping was low, we know that it is associated with the presence of p019 gene as also shown by other researchers. In this work po9 gene was liked only with kpc2 and kpc3 gene. The p019 gene has been reported to be present on other plasmids and mobile elements as well (10.1128/JCM.00238-21 ). Its interesting to describe this. There are two more publications that deserve inclusion as described below. Overall, I really liked the really liked the article and recommend for publication.
Rapid detection of KPC-producing Klebsiella pneumoniae in China based on MALDI-TOF MS
Yun Huang 1, Juan Li 1, Qianyu Wang, Kewen Tang, Congrong Li
https://doi.org/10.1016/j.mimet.2021.106385
A Full MALDI-Based Approach to Detect Plasmid-Encoded KPC-Producing Klebsiella pneumoniae
Miriam Cordovana1* Markus Kostrzewa2 Jörg Glandorf2 Michael Bienia3 Simone Ambretti1 Arthur B. Pranada
https://doi.org/10.3389/fmicb.2018.02854
Author Response
Thank you very much for taking the time to review and comment on our manuscript. Please, find the detailed responses below and the corresponding changes highlighted in the re-submitted file.
Reviewer's comment
This article is very interesting, very well written and works with a very important topic in clinical microbiology that is quick detection of KPC gene using MBT Subtyping Module. Although the overall sensitivity of the MALDI-ToF MS subtyping was low, we know that it is associated with the presence of p019 gene as also shown by other researchers. In this work po9 gene was liked only with kpc2 and kpc3 gene. The p019 gene has been reported to be present on other plasmids and mobile elements as well (10.1128/JCM.00238-21). It’s interesting to describe this. There are two more publications that deserve inclusion as described below. Overall, I really liked the really liked the article and recommend for publication. Rapid detection of KPC-producing Klebsiella pneumoniae in China based on MALDI-TOF MS Yun Huang 1, Juan Li 1, Qianyu Wang, Kewen Tang, Congrong Li https://doi.org/10.1016/j.mimet.2021.106385
A Full MALDI-Based Approach to Detect Plasmid-Encoded KPC-Producing Klebsiella pneumoniae Miriam Cordovana1* Markus Kostrzewa2 Jörg Glandorf2 Michael Bienia3 Simone Ambretti1 Arthur B. Pranada https://doi.org/10.3389/fmicb.2018.02854
- Response: Both references are included in the revised manuscript.
Reviewer 2 Report
Dear authors,
The article aims to test the performance of KPC detection by the MALDI Biotyper® MBT Subtyping Module using clinical Antibacterial Resistance Leadership Group (ARLG) Klebsiella isolates.
The article is well written, with good English level and scientific language. However, some points can be updated.
Questions:
1. What is the novelty of the work? Please clarify this aspect and include it in the article.
2. It Will be interesting to add a paragraph in the introduction section regarding the process associated with Maldi-Tof. How does it work to identify bacterial species?
Minor corrections
Line 68 – Add the citation (Lau et al., 2014).
Check references, some of the titles are totally in italic style
I recommend the authors add some more information regarding previous studies performed with Maldi-TOF in Klebsiella isolates. For example, you can add these studies https://pubmed.ncbi.nlm.nih.gov/34576808/ and https://journals.asm.org/doi/10.1128/jcm.01751-22
good English level
Author Response
Thank you very much for taking the time to review and comment on our manuscript. Please, find the detailed responses below and the corresponding changes highlighted in the re-submitted file.
|
Question |
Reviewer’s evaluation |
Response |
|
Does the introduction provide sufficient background and include all relevant references? |
Can be improved |
Done |
|
Are all the cited references relevant to the research? |
Can be improved |
Done, included other relevant citations |
|
Are the conclusions supported by the results? |
Can be improved |
Done, improved the contextualization of findings |
What is the novelty of the work? Please clarify this aspect and include it in the article.
- Response: Done.
It will be interesting to add a paragraph in the introduction section regarding the process associated with Maldi-Tof. How does it work to identify bacterial species?
- Response: Done.
Line 68 – Add the citation (Lau et al., 2014).
- Response: Done.
Check references, some of the titles are totally in italic style.
- Response: Done.
I recommend the authors add some more information regarding previous studies performed with Maldi-TOF in Klebsiella isolates. For example, you can add these studies https://pubmed.ncbi.nlm.nih.gov/34576808/ and https://journals.asm.org/doi/10.1128/jcm.01751-22
Response: Thank you for suggesting additional literature to strengthen the background of our paper. The suggested work by Gato and colleagues was included in the original submission. We did not include the study by Carvahlo and collaborators since this group only used MALDI-ToF for species identification of Klebsiella pneumoniae and Escherichia coli isolates, but not for detection of resistance mechanisms. Antimicrobial susceptibility testing was performed by disk diffusion and detection of genes confirmed by PCR and sequencing.